# Long-Term Seed Dispersal within an Asymmetric Lizard-Plant Interaction

**DOI:** 10.3390/ani13060973

**Published:** 2023-03-08

**Authors:** Ana Pérez-Cembranos, Valentín Pérez-Mellado

**Affiliations:** Department of Animal Biology, Universidad de Salamanca, Campus Miguel de Unamuno, 37071 Salamanca, Spain

**Keywords:** *Podarcis lilfordi*, Lacertidae, islands, Balearics, seed dispersal, endozoochory, mutualism

## Abstract

**Simple Summary:**

For 24 years, we studied the interaction of the endemic Balearic lizard *Podarcis lilfordi*) and the dead horse arum (*Helicodiceros*
*muscivorus*) on Aire Island, a small coastal islet of the Balearic Archipelago (Spain). From a small initial plant population, the frugivorous activity of lizards apparently resulted in an extraordinary increase in plant abundance over the entirety of Aire Island. The intensity of seed dispersal by lizards was higher in years with lower rainfall and, consequently, with lower availability of other food resources. Thus, the fruits of the dead horse arum were an alternative food strongly exploited during years of food shortage.

**Abstract:**

During the last 24 years, the mutualistic interaction between the dead horse arum, *Helicodiceros muscivorus*, and the Balearic lizard, *Podarcis lilfordi*, was studied on Aire Island (Balearic Islands, Spain). From a small population of a hundred plants, the dead horse arum expanded extraordinarily throughout the island, reaching the highest known densities of the species and occupying areas of the island where it was not previously present. The current abundance of plants is a direct effect of the frugivorous activity of the Balearic lizard, which is the main, if not the only, effective seed disperser of the plant on Aire Island. However, abiotic factors predominated over biotic factors in driving abundance of plants. Over the years, plant densities varied significantly depending on the aridity of the island, with higher densities recorded in drier years. Lizards’ frugivorous activity and dispersal intensity was inversely correlated with annual rainfall. We found higher dispersal intensity in years with lower rainfall. We propose that the years of lower rainfall are those in which there is a lower prey availability. In such years, lizards compensate the shortage of other trophic resources with a more intense consumption of dead horse arum fruits. The mutualistic interaction is therefore asymmetric, since there is a greater influence of the frugivorous activity of the lizards on the plants than of the plants on lizards. It is, in short, a system chronically out of balance.

## 1. Introduction

Islands are widely recognized as suitable places for the rise of interactions between plants and animals [1]. In several cases, such interactions are clearly mutualistic, resulting in mutual benefits for the organisms involved. Vertebrates may be essential agents in plant dispersal; about half of vascular plant species have vertebrate seed dispersers [2]. Among reptiles from insular ecosystems of the Mediterranean basin, frequent mutualistic interactions between lizards and vascular plants, which include phenomena of seed dispersal, have been identified [3,4,5,6,7,8]. An indirect consequence of these interactions is that plants are more vulnerable to climate change because of the disappearance of dispersing fauna, especially vertebrates ([9,10] (and references therein)).

Mediterranean plants exhibit several characteristics associated with seed dispersal by animals to a greater extent than plants from temperate habitats, such as fleshy fruits or seeds bearing elaiosomes [9]. Consequently, the probability of dispersal relationships between lizards and plants are higher in the Mediterranean basin. As classic examples of mutualism, frugivory and seed dispersal are widely studied in a large variety of vertebrates and plant species. All major lineages of vertebrates take part in fruit consumption and seed dispersal [10], but lizards are only prominent in arid and insular environments [5,7,8]. In addition, the earlier notion of lizards as specialized frugivores relying almost exclusively on fruits has been challenged, and today it is clear that most of them also include prey in their diets.

The dead horse arum, *Helicodiceros muscivorus* (Engler, 1879), is a vascular geophyte plant of the Araceae family (Aroideae) restricted to islands of the western Mediterranean, including Corsica, Sardinia, Mallorca, and Menorca islands [11,12]. It is a paleoendemic, originating in the western Mediterranean from ancestors with a very wide distribution during the early Oligocene, 31.1 ± 6.7 million years ago [13]. However, other authors consider that its origin is more recent, in the middle Miocene, about 12–13 million years ago [14]. Therefore, the dead horse arum is a relictual species originating from the so-called Hercynian Islands, with phylogenetic affinities with the Araceae of the central and eastern Mediterranean Basin [15,16]. The dead horse arum would have originated thanks to a vicariance process and not to dispersal of its predecessor. This hypothesis seems logical, given the low dispersal capacity of its fruits [14].

The Balearic lizard, *Podarcis lilfordi* (Günther, 1874), is an endemic lacertid lizard inhabiting the Balearic Islands. It is an active forager that captures insects and other invertebrates, but it also consumes vegetal matter, carrion, conspecifics, or leftovers deposited by tourists [17].

Over several years, we studied the frugivorous behaviour and seed dispersal of the Balearic lizard (*Podarcis lilfordi*) on Aire Island (Menorca, Balearic Islands, Spain) as a part of the complex interaction between the Balearic lizard and the dead horse arum [18,19]. In this study, we present the results of long-term research on seed dispersal during a period of 24 years. Our first seed dispersal studies, carried out in 1997 and 2003, indicated that the intensity of dispersal was greater in areas of higher plant density. In addition, we observed that some years lizards seemed to select, among those available, the largest fruits [18], a result that was not confirmed later with bigger sample sizes [19,20]. In addition, the expansion of the distribution of the plant on the island made it possible to study seed dispersal over its entire area. Our preliminary hypothesis is that dispersal intensity would be directly related to some biotic factors, such as annual plant abundance, as well as to the abundance of the main seed disperser (i.e., the Balearic lizard), as well as to abiotic factors, such as the annual rainfall. We could expect that dispersal intensity would be higher during years with higher rainfall, higher plant abundance, and/or higher density of lizards.

## 2. Materials and Methods

### 2.1. Study Period and Area

The study was carried out on Aire Island during spring and summer, from 1997 to 2022. Aire Island (Figure 1A) is located off the south-eastern coast of Menorca (Balearic Islands, Spain). The island has a surface area of around 34.4 hectares. It is a flat island, with a slow gradient rising to 15 m above sea level along a north to south axis. In the central part of the northern coast, there is a jetty suitable for dockings of small to medium boats. The buildings on the island are all in ruin, except for a small hut by the jetty and the lighthouse, now automatic, which was maintained by a resident lighthouse keeper up until 1975.

Aire Island has a mesomediterranean climate, characterised by mild temperatures with very seasonal rainfall, abundant rains in winter, and very dry summers. There are frequent and strong winds, principally from the north and the east. Aire Island is in the driest part of Menorca [21]. Between 1945 and 1975, an average annual rainfall of 435 mm was recorded. Annual average temperature is 17.2 °C, with a maximum of 27.5 °C in July and a minimum of 9.5 °C in January (there is a weather station on Sant Lluís, the closest to Aire Island, [21]).

Regarding vegetation, due to its flatness and its relative distance from the coast, the whole surface of the island is subjected to the influence of the sea. Consequently, except for communities of shallow rooting plants, the vegetation is mainly hallophyllous. Typical of places strongly affected by the sea, species, such as *Crithmum maritimum* and *Limonium* spp., can be found. A little further from the coast, the vegetation opens out into shrub communities dominated by *Suaeda vera*, a typical species of saline soils. There are plantations of *Tamarix africana*, as well as typical Mediterranean species, such as *Pistacia lentiscus* and *Phillyrea media*. Furthermore, *Carlina corymbosa* forms dense patches in the least saline areas (Figure 1B).

For our seed dispersal study, we consider five different areas of Aire Island, differing in abundance of the dead horse arum (Figure 1C and Figure 2, and see [18]): the central area of the island, with its southern part showing the highest plant densities (the so-called high-density area, H.D. from Figure 1C and Figure 2) and the northern zone with lower plant densities (low-density area, L.D. from Figure 1C and Figure 2); the northern peninsula (N.P. from Figure 1C and Figure 2), with sandy substrates; the eastern area (E.A., Figure 1C), with rocky susbtrates; finally, the western peninsula (W.P. from Figure 1C and Figure 2), occupied every year by a large breeding colony of yellow-legged gulls, *Larus michahellis*, and a reduced colony of the Audouin’s gull, *Ichtyaetus audouinii*, that can breed some years on eastern area of the island. The western peninsula had very low abundance of dead horse arum some 30 years ago, but its density has increased remarkably in later years (see below). We have much more information from the high-density area (H.D., Figure 1C and Figure 2) in the center of the island. In the H.D. area, we obtained estimates of plant densities for 19 years, while the intensity of dispersal has been estimated for 20 years, as well as the density of lizards (Figure 1 and Table A2).

### 2.2. Study System

#### 2.2.1. The Lizard

*Podarcis lilfordi* (Squamata, Lacertidae) is a medium-sized lizard (Figure 1D) with a maximum snout-vent length of 81 mm in males and 75 mm in females. It inhabits the coastal islets associated with Menorca, Mallorca, and those comprising the Cabrera archipelago [22]. The Balearic lizard is an endemic and endangered species of the Balearic Islands and one of the three species of terrestrial vertebrates from the pre-human Plio-Pleistocenic fauna of the archipelago [21]. This lizard reaches high densities in Aire Island [23].

#### 2.2.2. The Plant

The dead horse arum, *Helicodiceros muscivorus* Engler 1879 (Araceae, Aroideae), is a plant species with an extraordinary deceptive pollination system, imitating a carcass of a mammal or bird by means of visual, olfactory, and thermal cues, attracting blowflies (Diptera, Calliphoridae), which are then employed as unrewarded pollinators [24]. During the blooming period, the plant produces an intense odor of decaying meat and attracts female blowflies. Flies arrive at the plant and enter the floral chamber through a tubule that likely simulates a natural orifice of a dead animal. Flies that enter are trapped in the floral chamber where, due to their continuous attempts to escape, they transfer previously loaded pollen grains from another plant to receptive female flowers [24] (and references therein).

At the end of spring, on variable dates depending on the year and the weather conditions, the fruiting of the dead horse arum begins. The infructescence (Figure 1E) is formed by closely grouped berries arranged around a floral axis. Fruits are ovoid, with an average of maximal width of 9.4 mm and an almost liquid pulp and shades that range from green and whitish to intense orange or red, depending on the ripening state and each individual fruit [18]. The seeds are also ovoid, generally rough, and brown in color, with an approximately conical elaiosome or strophyll, which is also brown in color [25]. Fruits of the dead horse arum can contain from 1 to 8 seeds of around of 2.5 to 3 mm of maximal length and 0.2 g of weight [18]. Until the discovery of the consumption of the fruits by the lizards of Aire Island, the only reference to the dispersal of this plant came from data obtained in Cabrera Island, where the Balearic lizard acts as a seed disperser [1,4,26].

### 2.3. Distribution and Abundance of Plants and Lizards

In the heart of the high-density area (H.D.), to the east of the path that leads from the dock to the lighthouse, we established four line transects that are 25 m long and two meters wide, orienting the four lines towards the four cardinal points (Figure 1C). These four transects have been replicated, from 1999, over 19 years (Table A1). In the rest of the areas described above, the densities of lizards and plants have been estimated by means of line transects that are 70 m long; plants were censussed within 2 m of both sides of the line transect and lizards within 4 m. These four transects of plants, replicated much more frequently over the years, are used for interannual comparisons, together with estimations of lizard densities performed at high-density (H.D.) and low-density (L.D.) areas (Figure 1C and Table A2).

Density estimates of plants and lizards (see a general methodology in [27]) have been carried out in the R environment [28]. Field counts of lizards and plants were performed by the second author during the first years of study and then, from 2009, by the two authors. During the first years of study, densities were estimated with the package Distance (ver. 6.0 and earlier; [23]). Since then, densities have been estimated with N-mixture models using the “unmarked” package within R environment. N-mixture models can provide accurate abundance estimations of small vertebrates [29]. Unmarked is a unified modelling framework for hierarchical models [30]. We used the model of abundance from distance sampling with the fitting function “distsamp” that fits the multinomial Poisson model to distance sampling data [31]. The probability of detection for lizard densities was modeled as a function of the distance (d) to the observer using the half-normal detection function [30]. For plant densities, we employed the uniform function, since there is an equiprobability of detection throughout the width of the one-meter band on each side of the transect line. For each analysis, the density in number of individuals per hectare ± the standard error (SE) is provided, as well as the number of contacts and the length of the line transects in meters. For lizard transects, we also calculated the value of the probability of detection, g(x), given by “unmarked”.

To establish the present-day distribution of the dead horse arum, we made an abundance map of plants, completed with data from 2015 to 2022. The map has been prepared in Google Maps^®^ with four levels of abundance, indicated by four color intensities (Figure 2). To establish the range limits of plants, data from the most recent years have been used to estimate densities, that is, the information obtained in 2021 and 2022. In addition, from 2019 to 2022, we searched for occurrences of the plant in the western peninsula (W.P.) and in the easternmost part of the island (E.A.).

### 2.4. Seed Dispersal

Fruits and seeds were sampled during the months of May, June, July, and August, covering the entire fruiting period of the dead horse arum on Aire Island [18]. However, seed dispersal analyses used the highest seed dispersal values that were always observed during May or June. Since the high-density area has been the most intensively sampled, we focused in that area our long-term comparisons. We obtained data from four different areas of the island called: high-density (H.D.) of plants area, low-density (L.D.) area, northern peninsula (N.P.), and western peninsula (W.P., Figure 1C and Figure 2).

Seed dispersal intensity, measured as the percentage of faeces containing seeds from *Helicodiceros muscivorus*, was intermittently studied from 1997 to 2022, over a total of 19 years, from which comparable data were available. We do not have reliable data of seed dispersal intensity from 2020 because, due to the COVID-19 pandemic, we arrived too late (late June) to the study area. We were interested in establishing whether seed dispersal was related to the abundance of plants each year, to lizard abundance, as the main seed dispersers [18], or to abiotic factors, such as annual climatic conditions.

Every year we attempted, at the minimum, one seed dispersal sampling at the end of the fruiting period, in the month of June. However, sometimes, it was possible to carry out weekly samplings or several samplings throughout the whole fruiting period. In our analyses, we always employed the maximum dispersal intensity recorded. The usual sample size was 50 faeces per area, but in some cases, it was lower or higher (see Results and Table A3). For each faecal sample, we determined the presence and number of *Helicodiceros muscivorus* seeds, including seed measurements, and then we recorded these data in a database to make comparisons with seed numbers and sizes per fruit obtained directly from plants. These data were employed in other studies (Pérez-Cembranos and Pérez-Mellado, in preparation).

### 2.5. Climatic Information

For all years under study, the values of monthly rainfall, total rainfall, monthly temperatures, relative humidity, and atmospheric pressure were obtained from public data provided by AEMET (Spanish Meteorological Agency, [32]). Data came from the weather station of Menorca airport. With this information, we calculated the annual rainfall, as well as what we called “valid” rainfall, that is, the total precipitation of 12 months prior to April of a given year, April being the main blooming period of *H. muscivorus* in Aire Island. We consider that, during this period, from May of the previous year to April of the current year, the rainfall with the greatest influence on primary productivity occurs and, therefore, it affects plant and animal productivity. Likewise, for valid months, we calculated two climatic indices: Iq index [33] and Emberger’s Q index [34].

### 2.6. Statistical Analyses

Due to its non-normal distribution, plant densities were compared with annual climatic variables using Spearman non-parametric rank correlations. To analyse annual variation of dispersal intensity, we employed a multiple regression within the “base” package of R, with highest values of dispersal intensity in high-density area as the response variable and three different continuous explanatory variables: the valid rainfall of the year, the density (individuals/hectare) of plants, and the lizard density (individuals/hectare) in that area. In the maximal model, we included these three lower linear terms, as well as associated quadratic terms and all interactions of the three explanatory variables. From this maximal model, we manually removed, step by step, non-significant interactions and quadratic terms. We used ANOVA analysis to compare models produced by stepwise deletions. Finally, we obtained the minimum adequate model [35]. Because the assumptions of the multiple regression were only partially met (see below), we also ran a model of quantile regression with the three explanatory variables. Additionally, to show the relation of dispersal intensity with an explanatory variable, we calculated the product–moment correlation between these variables, and we made a graphical exploration of correlation among variables. All analyses were performed within the R environment [28]. Quantile regression was performed with the package “quantreg” [36,37].

## 3. Results

### 3.1. Plant and Lizard Densities

The dead horse arum is today present practically throughout the whole Aire Island area, although with different densities in each zone (Figure 2). Four levels of abundance have been defined: the first of low abundance (approximately 1100 individuals/hectare); the second of medium abundance (about 5900 individuals/ha); the third of high abundance (around 16,650 individuals/ha); and the fourth with very high abundance (about 36,300 individuals/ha, Figure 2). The high-density area (H.D., Figure 2), mainly covered with the shrubby sea-blite, *Suaeda vera*, is the area with highest recorded densities of dead horse arums, while the plant is less common in the low-density area (L.D., Figure 2). Through the last 20 years, an extraordinary increase in plant densities took place (Figure 3 and Table A1), particularly between 1999 and 2015, as well as an expansion in areas of the island where dead horse arums were absent or very scarce at the end of the 20th century. For example, in 1998, the existence of only some 100 young plants was recorded [38]. During our first studies [18,19,20], the northern peninsula (N.P., Figure 2 and Table A2) was considered an area without *Helicodiceros muscivorus*, as well as with very few plants in the western peninsula (W.P., Figure 2). At least since 2003, the dead horse arum has spread to all areas of Aire Island with topsoil and some rocky or shrub cover and protection [18].

Among the areas of Aire Island, there are significant differences in dead horse arum densities (Kruskal-Wallis’ test, χ^2^ = 15.083, *p* = 0.019, Figure 4), with higher densities in the central part of the island (east part of the high-density area, Figure 4). Plants are less common in the northern peninsula and the eastern part of the island (Figure 4 and Table A2).

Over the years of study, for the high-density area (H.D.), from which we have the best sampling of plant densities (four lines transects of 25 m), there is a significant correlation of plant density with Emberger’s aridity index (Spearman rank correlation, S = 1886.3, *p* = 0.0023, d.f. = 17; ρ = 0.6546) or with Iq aridity index (S = 1816.3, *p* = 0.0074, d.f. = 17; ρ = −0.5932, Figure 3). Thus, plant densities were higher in more arid years. However, if we only consider the valid rainfall, we did not find any significant correlation with plant densities in the high-density (H.D.) area (S = 1272.1, *p* = 0.63, d.f. = 17; ρ = −0.116).

### 3.2. Seed Dispersal

Since the first study in 1997, it became clear that seed dispersal influenced plant abundance. In fact, between 1999 and 2005, an extraordinary increase in plant density was observed in the high-density area (H.D., Figure 3 and Table A1 and Table A2).

Dispersal intensities offer different pictures at each area of Aire Island. Frugivorous activity of lizards and, consequently, dispersal intensity, was the highest in high-density area (H.D., mean = 61.7 ± 4.05%, n = 20 year; range: 26.67–92.31%), with lower annual average of intensity in low-density area (L.D., mean = 28.99 ± 3.46%, n = 19 years, range: 2–56%), northern peninsula (N.P., mean = 22.77 ± 5.82%, n = 9 years, range: 0–56%) or western peninsula (W.P., mean = 40.08 ± 10.64%, n = 9 years, range: 3–80%, Figure 5). Between the two areas (L.D. and H.D.) for which we have 18 comparable years, dispersal intensity was significantly higher in high-density (H.D.) area (Mann-Whitney paired test, V = 171, *p* = 0.00021).

If we consider the high-density area (H.D.), we can explore which factors could be related to dispersal intensity. From the graphical exploration of correlations among response and potential explanatory variables, we performed multiple regression analysis. After manual simplification, the minimum adequate model excluded the three quadratic terms, as well as all interactions. This final model included the three initial explanatory variables, showing that only the valid rainfall had a significant influence on dispersal intensity (t = −5.102, *p* = 0.002, multiple R^2^ = 0.689, R^2^ = 0.617, Figure 6). Thus, we were unable to detect any significant effect of lizard densities on dispersal intensity (t = 1.1129, d.f. = 17, *p* = 0.2812, ρ = 0.2606, Figure 7) or any relation with plant densities (t = 0.5076, d.f. = 16, *p* = 0.6186, ρ = 0.1259). The negative correlation between dispersal intensity and valid rainfall (Pearson’s product–moment correlation, t = −3.3389, d.f. = 18, *p* = 0.0036, ρ = −0.6184, Figure 8) indicates that, with lower values of rainfall, we obtained higher dispersal intensities. Results from the quantile regression analysis are similar, and, in all cases (for ρ = 0.25, 0.50 and 0.75), valid rainfall was the only significant explanatory variable (ρ = 0.25, t = −3.816; *p* = 0.002; ρ = 0.50, t = −4.429, *p* = 0.0007; ρ = 0.75, t = −5.103, *p* = 0.0002).

Finally, dispersal can have a direct influence on the subsequent dead horse arum density. This relationship does not appear with the previous year’s dispersal intensity, but there is a statistically significant correlation between the density of *Helicodiceros muscivorus* and the intensity of its dispersal by *Podarcis lilfordi* two years earlier (Pearson’s correlation, t = 2.4013, d.f. = 13, *p* = 0.03201; correlation = 0.5543; Figure 9). This result implies that the frugivorous activity of the lizards itself has a direct, although not immediate effect, on the abundance of plants, at least in the high-density area.

## 4. Discussion

The dead horse arum is, during spring and the first half of the summer, a very abundant plant on Aire Island. The plant is very conspicuous during its blooming and fruiting periods (Figure 10). If we compare Aire Island with the other localities where *Helicodiceros muscivorus* is present at the Western Mediterranean Basin, we must conclude that the abundance of the dead horse arum in Aire Island is, probably, the highest in the world, far greater than the abundancies in the known nuclei of Mallorca and Menorca, as well as Corsica and Sardinia [11,12]. For example, in high-density area (H.D.), we recorded more than six individual plants per m^2^ (Table A1). This fact, by itself, identifies Aire Island as a key role as the home of the most important population of this Tyrrhenian endemic plant.

Our first observations in the 1990s of the 20th century [17,18] seem to indicate that the plant was restricted in its distribution to the central zone of the island (high-density area (H.D.), Figure 2), within the extensive stands of *Suaeda vera* shrubs. We lack information on the distribution of the plant in the second half of the 19th century and the beginning of the 20th century, when it was mentioned by Rodríguez Femenías [39,40]. In this period, the plant coexisted on Aire Island with a constant human presence, the lighthouse keepers. This presence was accompanied by domestic livestock, especially goats. Moreover, we do not know the size of the population of the Balearic lizard in this period. Probably, it was large, as is generally the case on the coastal islets of Menorca and Mallorca [23,41,42], although disturbed to a greater or lesser extent by the presence of humans, domestic livestock, and terrestrial predators of lizards, such as cats and dogs.

In such a scenario, the interaction between the lizards and the vascular plants of Aire Island would perhaps be less intense than it is today, as in the case of larger islands, where there is a constant human presence. It is well known that the absence of predation pressure and, in general, of human pressure, is usually the case in populations where greater interactions between plants and lizards are observed and where there are other natural history traits typical of undisturbed populations [7,42] (and unpublished data). The consumption of nutritional elements from plants, such as nectar, pollen, and even fruits, is an activity that takes considerably longer time than the capture of fast prey. Therefore, the consumption of plant elements usually appears in populations with reduced predation pressure, where the time of exposure to predators is not a limiting factor [42].

Based on these considerations, we can speculate that the expansion of the dead horse arum, enabled by the intense lizards’ extensive seed dispersal, only began after the automation of the lighthouse and the disappearance of domestic animals and permanent inhabitants. The dramatic increase in plant density observed around 20 years later could have been the result of interaction with lizards in an environment without terrestrial predators and without other disturbances due to human presence. In short, an ecological release would have been taken place, as in several insular populations. This is the general hypothesis to explain the existence of numerous aspects of island ecology that are absent or rare in continental ecosystems [43].

Seeds of *Helicodiceros muscivorus* carry elaiosomes, indicating a potential importance of ants as dispersal agents [44,45,46]. Probably, elaiosomes provide a reward to omnivorous lizards, in addition to nutritional compounds of ripening fruits. We already showed that the digestion of the elaiosomes by lizards significantly benefits seed germinability of the seeds [18]. During the fruiting period of *Helicodiceros muscivorus*, we observed that Balearic lizards strongly compete with some ants that also consume fruits [18]; in this case, the ant species is *Tapinoma nigerrima* (Nylander, 1856), a small Mediterranean ant, particularly abundant in areas with some degree of human alteration [47]. Frequent aggressive interactions between ants and lizards were observed during this period. Those interactions between ants and small reptiles have been frequently recorded, even interactions between myrmecophagous lizards and ants [48]. In these encounters, when the lizards are attacked by ants during their frugivorous activity, the lizards do not usually react by capturing ants, even though they are common prey items [48] (and references therein). This is also true of the Balearic lizard, usually a myrmecophagous species [22,49], which does not respond to attacks by ants on the infructescences, by capturing and consuming them. In addition, seed dispersal and predation data seem to indicate that the role of ants as seed dispersers is reduced on Aire Island [18]. However, it is true that, in general, there is little information on the dispersal capacity of ants in the Mediterranean Basin [50,51].

The fruits of the dead horse arum are rarely consumed by birds on Cabrera Island, south off Mallorca [26], and we never observed their consumption by birds on Aire Island. In the high-density area (H.D.) of Aire Island, the fruits of the dead horse arums are an important trophic and, probably, water resource for lizards during a narrow period of late spring. There are additional reasons for the net positive effect of dispersal by lizards. Unlike birds, lizards can easily move into rocky crevices and areas of dense vegetation, thus accessing infructescences inaccessible to birds [52]. Access to these microhabitats also allows the deposition of seed-bearing droppings in particularly well protected places for later germination [52,53,54]. In addition, it should be noted that the inclusion of the seed inside a dropping protects it from seed predators [18].

We observed that the density of *Helicodiceros muscivorus* undergoes considerable interannual variations, particularly evident in the central area of the island (H.D. amd L.D. areas), where we find the highest known densities of dead horse arum. In the high-density area (H.D.), we found a negative and statistically significant correlation between the annual density of dead horse arum and the aridity indices, Q, of Emberger and Iq. This result indicates that the abundance of plants has been higher in drier years. Adler and Levine [55] showed that, in the same North American grassland locality, the annual variations in precipitation prior to density estimation had a negative and statistically significant correlation with plant richness. Thus, the greatest richness corresponded to the driest years. These authors proposed several explanatory hypotheses for such an apparently unexpected result. The greater abundance of plants after a year of low rainfall could be related to the need for drier conditions to guarantee seed opening or to promote a greater mobilization of edaphic nutrients, such as nitrates. Furthermore, the positive effects of aridity were shown to be particularly relevant in annual plants [55]. In this sense, let us not forget that the seeds of *H. muscivorus*, as well as the buried tubers, are not very resistant to frost and humid winters [56]. Regardless of the explanation of the phenomenon, this result indicates that the requirements of each plant species can be highly variable and that, in the case of *H. muscivorus*, a greater aridity seems to be favorable to germination and survival of the buried tubers.

Dispersal intensity does not seem to bear any relation to the abundance of plants in the same year. However, the density of plants is apparently related to the intensity of their seed dispersal by lizards two years earlier. We do not know why the effect of dispersal intensity is only detected after two years. Seed germination may require two years for a significant fraction of seeds. Unfortunately, our knowledge of the reproductive biology of the dead horse arum is fragmentary, and there are no data on the rate of germination under natural conditions. Under controlled conditions, germination and visually detectable plant growth occurs in about three months [18].

Moreover, there is less frugivorous activity and dispersal in years with higher rainfall and more intense frugivory in years with low rainfall. Such a result seems counter-intuitive, but it may have an appropriate explanation if we consider that Balearic lizards are omnivorous ectotherms and consume, in addition to many other plant species, a large variety of prey, as well as marine subsidies, carrion, and even conspecifics [17].

In a comprehensive analysis of relationships of annual climatic changes and primary productivity, Maurer et al. [57] showed that rainfall had the highest explanatory power of productivity over annual aridity variations. The reason is that precipitation exhibits the highest interannual variability, over other abiotic factors, such as temperatures included in aridity indices. In fact, there is a well documented relation between annual precipitation and net primary production [57,58] (and references therein). That is particularly clear in the case of Mediterranean arid ecosystems [58,59].

In general, most terrestrial invertebrates are remarkably sensitive to environmental humidity since they have a high surface-to-volume ratio. Arthropods are usually scarcer during dry periods [60,61]. Thus, in dry years with reduced rainfall, many species will remain hidden, especially the epigean species. Arthropods with less sclerotized cuticles, such as terrestrial isopods, are important prey items in the diet of the Balearic wall lizard on Aire Island [17], and they will be unable to reduce water losses, and they will be even more vulnerable [62].

Global warming has led to the appearance of several studies [63,64] (and references therein) of the effect of precipitation on plant communities [63,65], as well as the effects on invertebrates and, particularly, on edaphic invertebrates [66,67] (and references therein). Presently, there is extensive experimental evidence of the effect of reduced soil moisture on trophic availability, oviposition behavior, survival, and abundance of soil fauna [68,69]. These studies have shown that the effects of low rainfall can be quite severe upon almost all groups of terrestrial invertebrates [67]. For example, terrestrial isopods are particularly sensitive to edaphic desiccation [70]. However, we must not forget that the effects of rainfall can be remarkably variable in some habitats and that generalizations are not easy on this topic [62]. In our case, as a tentative hypothesis, we can speculate that dry years will have a negative effect on the availability of invertebrates, both in primary and secondary consumers. That is, annual rainfall can be considered a reliable index of primary and secondary production [71]. Given this, dry years could be qualified as “bad” years in terms of prey availability for lizards. Precisely, in dry years, a greater intensity of frugivory and dispersal was observed on Aire Island.

Hence, consumption of dead horse arum fruits increases with lower availability of other food sources. This observation leads us to propose that dead horse arum fruits are an alternative, a secondary trophic resource in the feeding strategy of the Balearic lizard. On small islands, there seems to be a greater frequency of frugivorous strategies than on continents or large islands, which has led to the conclusion that it is the scarcity of other food sources that forces the observed plant consumption in insular lizards [72,73,74]. In addition, the nutritional value of the dead horse arum fruits is limited. The mean water content of pulp from fruits of *H. muscivorus* is very high (mean = 77.88 ± 0.46% of the total fruit weight [18]). This is characteristic of species whose fruits ripen during the dry summer, while winter-ripening plants show higher lipid contents [10,51,75]. In fact, even during the period of two to three weeks period of ripe dead horse arum fruit availability, the diet of the Balearic lizard includes a large fraction of arthropods [17], supplementing the poor protein and lipid intake associated with frugivory.

In short, the intense frugivory of the Balearic lizard on the dead horse arum is a strategy observed during a limited period, thanks to the ability of this species to track the seasonal phenology of several trophic resources [17], which are individually only available during a few weeks a year. Herrera [51] proposes a similar explanation for intensively frugivorous birds from the Mediterranean Basin.

The interaction between the dead horse arum and the Balearic lizard is a non-symbiotic and facultative mutualism. The two organisms can each survive without the presence of the other [1]. Frugivory of the Balearic lizard is important during the limited fruiting period of the dead horse arum. However, in an omnivorous species, such as *Podarcis lilfordi*, this fact apparently does not imply the existence of special digestive adaptations or a strict dependence of fruit abundance. This is also the case in insectivorous birds that exhibit intense frugivory during certain periods of the year [76]. In addition, we were unable to detect any selective effect of the intense frugivorous activity of the lizards on the dead horse arum fruit’s traits if we except for weak indication of the consumption of smaller fruits in high-density (H.D.) area (Pérez-Cembranos and Pérez-Mellado, in preparation).

The blooming period of the dead horse arum matches with the peak of seagull’s reproductive period in Aire Island. That is, the blooming probably coincides with the peak deposition of organic matter associated with seagull breeding colony as a key site for pollination of the dead horse arum [24]. With a blooming period around mid-spring, the ripening period can be the direct consequence of the peak in guano deposition, rather than a fruiting phenology adapted to the Balearic lizard as its seed disperser.

According to these results, the frugivory and seed dispersal of the Balearic lizard on dead horse arum and consequent seed dispersal is apparently in a chronic imbalance in the sense proposed by Herrera [77]. This author pointed out that most of the dominant plant species in the shrubby vegetation of the Mediterranean Basin that are dispersed by vertebrates belong to the group that he calls “ancient plants” [77,78]. These are species with a phylogenetic origin prior to the origin of present-day ecological conditions of the Mediterranean Basin. Hence, the production of fleshy fruits would be a basal trait in these plants species and, therefore, emerged before the current dispersal systems in the western Mediterranean [77]. It is obvious that *Helicodiceros muscivorus* is hypothesized to have a remote origin, dating back to the early Oligocene (more than 31 million years [13]) or, at the very least, to the middle Miocene (about 12 to 13 million years [14]). Thus, the dead horse arum would be in that group of ancient plants with an origin prior to the appearance of the ecological conditions that define the Mediterranean environment, some 3.2 million years ago [79].

We are in the presence of a very effective mutualism. However, it does not require mutual adjustments between the two species involved. In the case of the dead horse arum, the dispersal system has been particularly useful thanks to the omnivory of the Balearic lizard, which facilitates the continuous inclusion of new nutritional elements in its diet. This fact implies that any adaptive hypothesis in such an interaction should be tested by focusing our attention on the disperser, rather than on the plant. In short, one of the aspects that has been revealed as most important in the studies of these interactions in the Mediterranean Basin is that evolutionary adaptation is limited by the restrictions imposed by regional history and the phylogeny of the interacting species [51,80,81,82]. The indirect evidence that the interaction system of *Podarcis lilfordi* and *Helicodiceros muscivorus* constantly deviates from an equilibrium situation is the higher influence of abiotic factors, such as rainfall on plant abundance, the important temporal variability, and the weak reciprocal dependence of lizards and plants [10,78].

Although entering the field of pure speculation, we cannot rule out that the interaction between the Balearic lizard and the dead horse arum does not have an origin remote in time. We do not know if, in the ecological conditions of the Balearic Islands after the Messinian crisis, lizards, today extinct on the main islands of Mallorca and Menorca, had the opportunity to coexist there with the dead horse arum. If that was the case, the interaction could have arisen in a very remote period of the evolutionary history of both species to later disappear with the extinction of *Podarcis lilfordi* in Mallorca and Menorca main islands.

## 5. Conclusions

Even if plant–animal seed dispersal systems are characterized by the absence of obligate partnership and a weak mutual dependence between animals and plants [10], the case of the dead horse arum and the Balearic lizard is surprising in the sense of its strong effect on lizard’s foraging behavior, on plant abundance, and on its distribution in Aire Island. A reduced population of the dead horse arum can be maintained without any interaction with lizards, as is observed on other coastal islets of Menorca (Pérez-Cembranos and Pérez-Mellado, unpublished data). However, the emergence of the strong interaction on Aire Island appears to be the strongest explanation for the extraordinary density reached by plants at this locality. Even on Aire Island, with an apparent heavy dependence of the dead horse arum upon lizards, this relationship is clearly asymmetrical in the sense of Jordano [83]. The Balearic lizard is also the main disperser of several plant species [4] (and unpublished data).

## Figures and Tables

**Figure 1 animals-13-00973-f001:**
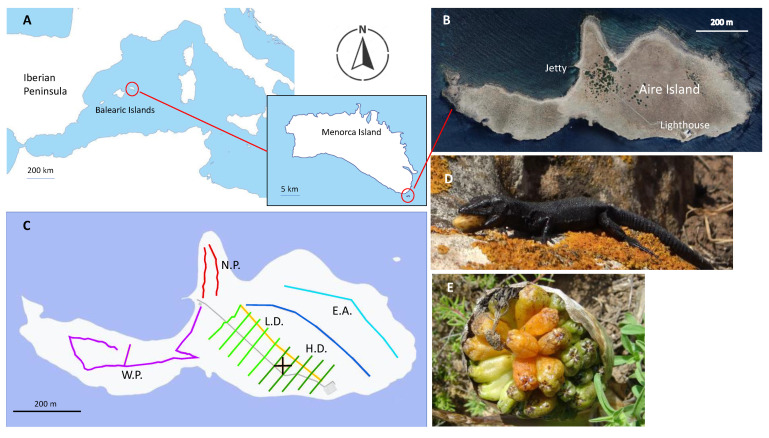
(**A**) Location of Aire Island off the southeastern coast of Menorca (Balearic Islands, Spain). (**B**) Orthophoto of Aire Island. Situation of the jetty and the lighthouse at either end of a main track (grey). (**C**) Location of lizard and plant line transects at different areas of Aire Island, Northern Peninsula (N.P., four line transects, red lines), Eastern Area (E.A., 12 line transects: six from inner east, dark blue lines, and six from coastal east, light blue lines), Western Peninsula (W.P., 12 line transects, purple lines), Low-density area (L.D., 10 line transects, light green lines, made separately to the east and to the west of the main track), and high-density area (H.D., 10 line transects, dark green lines, also separately to the east and to the west of the main track). In high-density areas, the black cross shows the position of the four line transects of 25 m of plant density sampled from 1999 (see more details in the text and Table A2). Additionally, every year we estimated lizard densities in low-density and high-density areas with line transects covering the central part of the island and ending at the enclosure of the lighthouse (yellow line). (**D**) An adult male of *Podarcis lilfordi* handling a ripe fruit of *Helicodiceros muscivorus*. (**E**) Top of an infrutescence of *Helicodiceros muscivorus* with ripe (orange) and unripe (green) fruits.

**Figure 2 animals-13-00973-f002:**
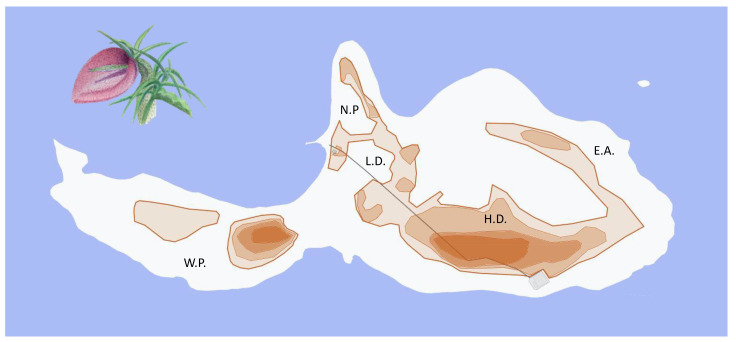
Distribution and densities of *Helicodiceros muscivorus* on Aire Island. This distribution is based upon data from 2021 and 2022.

**Figure 3 animals-13-00973-f003:**
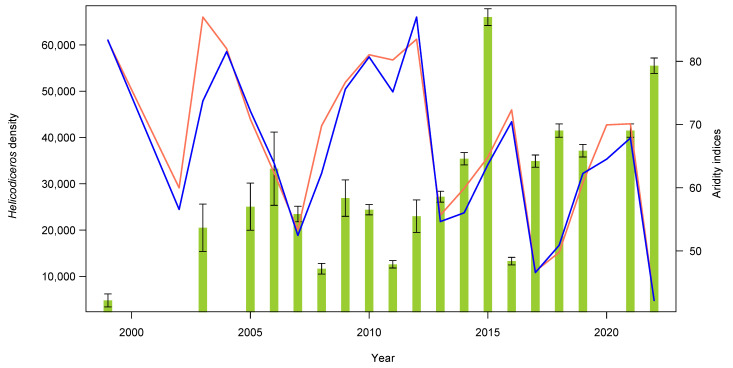
Annual density estimations of *Helicodiceros muscivorus* in high-density area (H.D.) of Aire Island. Left *y*-axis: green bars correspond to *H. muscivorus* density (individuals/ha ± SE) from 1999 to 2022. Right *y*-axis: values of Emberger’s aridity index (blue line) and Iq index (red line, see more details in the text). The highest plant density is recorded in 2015, with more than 60,000 plants/hectare.

**Figure 4 animals-13-00973-f004:**
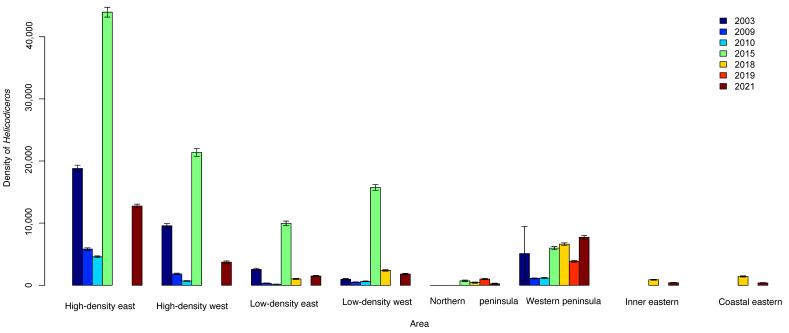
Dead horse arum density estimated (individuals/hectare ± SE) from seven years for different areas of Aire Island. Densities were separately estimated for eastern and western portions (east or west from the main track of the island) of high-density (H.D.) and low-density (L.D.) areas, as well as for the eastern area of the island (see also Figure 2 and more details in the text).

**Figure 5 animals-13-00973-f005:**
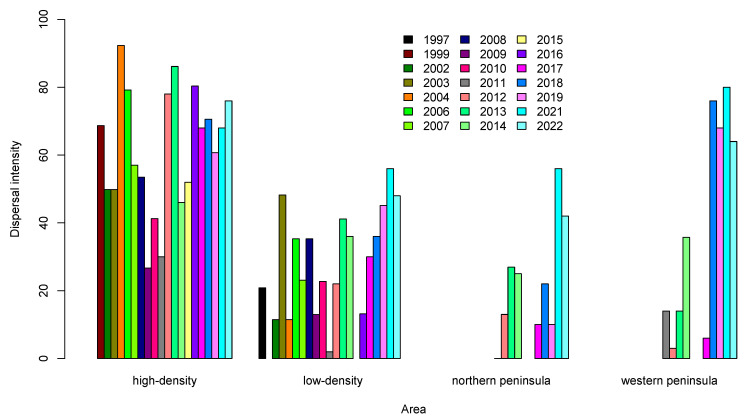
Dispersal intensities by the Balearic lizard in the four main areas of Aire Island during the entire study period (1999–2022, see more details in the text).

**Figure 6 animals-13-00973-f006:**
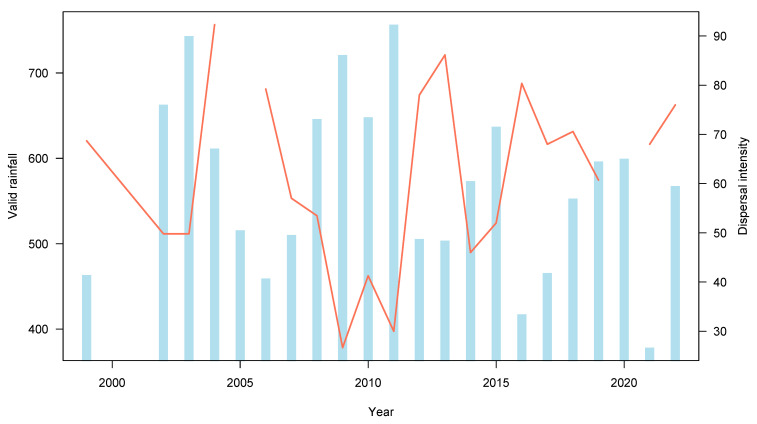
The relation of dispersal intensity of the dead horse arum (red line, right *y*-axis) and valid rainfall (blue bars, left *y*-axis) from 1999 to 2022 in high-density area (H.D.) of Aire Island.

**Figure 7 animals-13-00973-f007:**
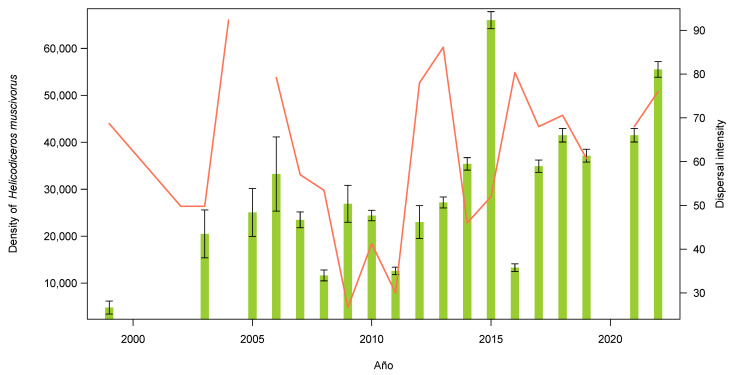
Annual dispersal intensity (red lines, right *y*-axis) and annual density of *Helicodiceros muscivorus* (green bars, individuals/hectare ± SE, left *y*-axis) in high-density area. The extraordinary increase in plant density from 2003 is clearly evident.

**Figure 8 animals-13-00973-f008:**
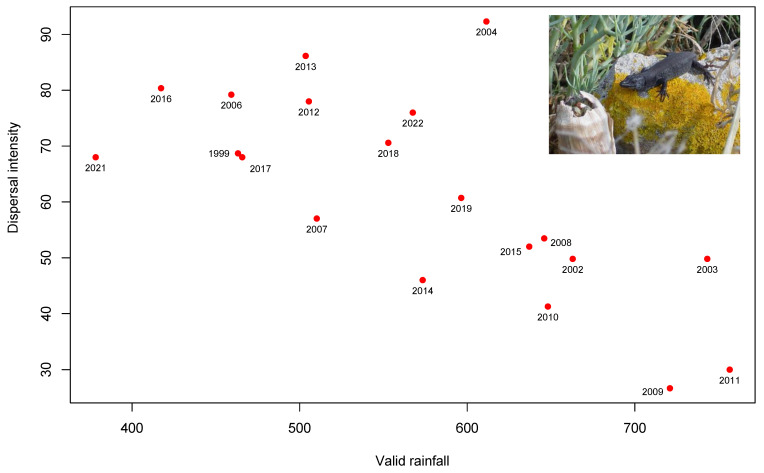
A negative correlation of dispersal intensity and valid rainfall at high-density area (H.D.) of Aire Island. In the upper right corner, an adult male of *Podarcis lilfordi* is looking at *Helicodiceros muscivorus* infrutescence, still with unripe fruits.

**Figure 9 animals-13-00973-f009:**
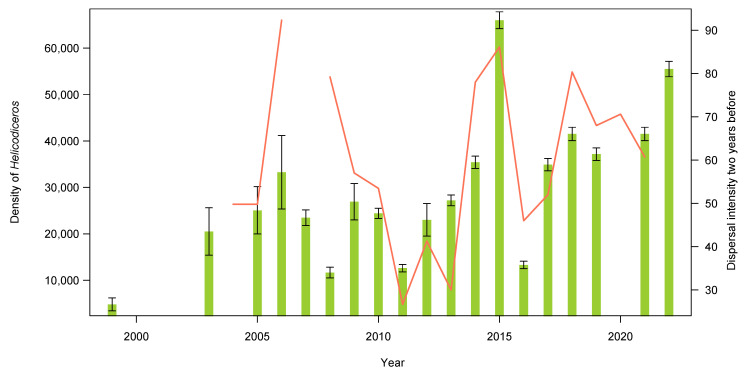
The relationship between dispersal intensity of the dead horse arum from two years previously (red lines, right *y*-axis), and plant density (green bars, left *y*-axis) (see more details in the text).

**Figure 10 animals-13-00973-f010:**
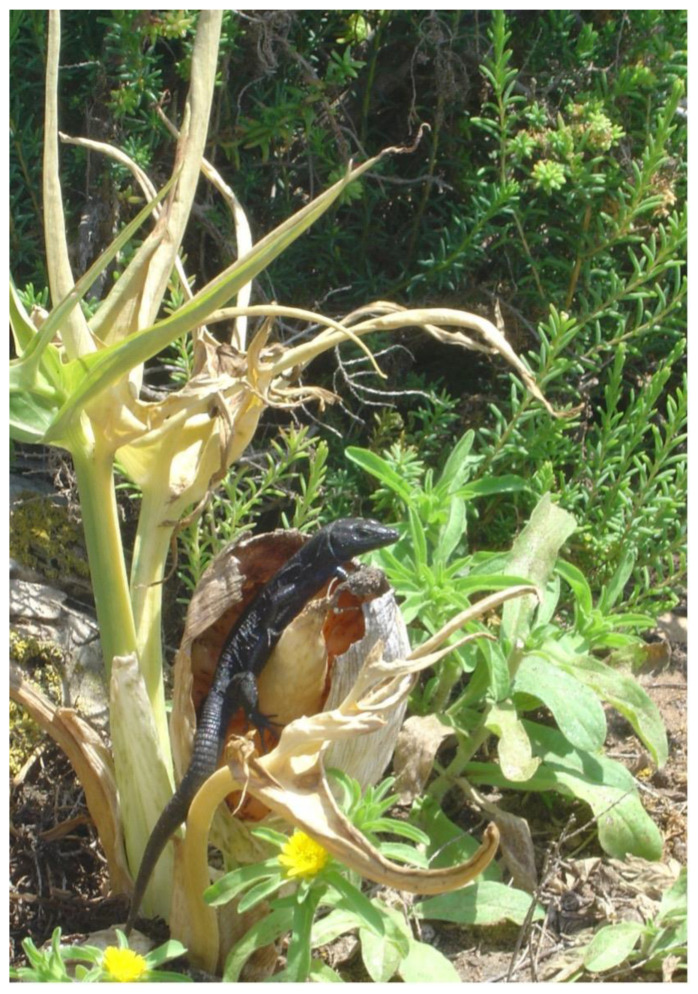
An intensely exploited infrutescence of dead horse arum with ripe fruits is visited by a female of the Balearic lizard.

## Data Availability

The data presented in this study are available on request from the corresponding author.

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
