# Peer review of "Long-Term Seed Dispersal within an Asymmetric Lizard-Plant Interaction"

_animals, 2023, doi:10.3390/ani13060973_

Round 1

Reviewer 1 Report

This is a very interesting system and I enjoyed reading about it. We don’t know enough about lizard-plant mutualisms and this study helps alleviate that deficiency. In addition, the lizard species is endangered, and the more we know about it, the better its chances of survival. I am recommending some minor changes.

The attached Word file has a lot of marginal comments on it. These are suggestions for ways of phrasing things, relatively minor questions, and suggestions. These constitute the bulk of my review. 

Some general points first:

·         The attached Word file consists of an annotated copy of your ms – I find Adobe worse than useless for editing and commenting. This resulted in changes in the formatting of the ms – I hope that this is not confusing. Also, the software I used to convert the PDF made a mess of converting all of the figures from Fig. 5 onwards, but I was able to look at those in the PDF and have nothing to say about them.

·         The first time a Latin binomial appears in a paragraph, the genus name should be written out in full. This is also true for figure and table captions. I tried to correct this throughout the manuscript but may have missed some instances.

·         You denoted the high-density area as “H.D.” and the low-density area as “L.D.”. However, you don’t use these contractions much elsewhere in the paper, instead referring to “high-density area” and “low-density area”. It would be less confusing for the reader if you used the contractions throughout the ms – you also refer to estimates of lizard and plant density (high and low), and it’s easy to confuse these with the labelled areas.

·         This lizard species is evidently extinct over much of its former range, due to human activity. Given that, a few words on its prospects and any conservation efforts made on its behalf would be apposite, perhaps in the Introduction.

The writing is clear and generally easy to understand, although you may think that I thought otherwise when you look at the emendations on the accompanying Word document. These are generally intended to phrase things as a native English speaker would put them – I hope that I haven’t changed the meanings of any of your text, and that you find these useful. There were a few places where I was not sure what you intended to say and have queried you in a marginal note. However, I generally found this ms easy to read.

If the editor requires you to respond to this review, please copy the marginal comments to which you are responding, with your responses, so I will be able to identify the point in the text more easily.

I am looking forward to seeing this published.

Author Response

Referee #1

[ N3]: Yes, we include the [8] reference, as a recent account of the topic.

[N5]: We agree with the comment of the editor, but we think that it is not necessary a deeper clarification of this point here because later in the text we showed that this lizard species is strongly omnivorous.

[N6]: The sentence can be referred to any frugivore among vertebrates, but in this case, we are talking about lizards. We modify the sentence.

[N7]: Really, it is a geophyte plant species. We clarify the sentence.

[N8]: Yes, we are speaking about annual plant abundance. We modify the sentence.

[N9]: Yes, Tamarix africana was deliberately plantated by humans in the islet.

[N10]: OK. We refer to these areas as L.D. and H.D. in the rest of the text.

[N11]: OK. We clarify this point.

[N12]: OK. You are right. We redraw Figure 1 considering all suggestions.

[N13]: Yes, we also changed these colors in the figure, as well as the heading of figure 1.

[N14]: Plant line transects of 25 m are fully explained in section 2.3.

[N15]: Ok. We include the conservation status of the species.

[N16]: OK. We move some contents to the Introduction.

[N17]: We do not know the taste of H. muscivorus fruits. But it seems that it could be quite bitter, because they are avoided by birds and the almost liquid pulp is a little bit irritating on the skin. The elaiosome is in the wider extreme of the seed. We include some information about the size of the fruits and the seeds from a previous study [17].

[N18]: OK. Done

[N19]: Yes. We include the two letters.

[N20]: We clarify the sentence about the line transect.

[N21]: OK.

[N24]: We include that it is per fruit.

[N25]: We clarify that we employed the “base” R package for multiple regression analyses.

[N26]: We observed a presence of plants in areas where they were previously absent and then, also an increase in plant abundance (density).

[N27]: I do not why you have this problem. We checked the pdf file of this Figure 4 and colour codes are present. We send again that figure in pdf format. In our Word version of the manuscript, it is also possible to see these codes.

[N28]: Yes.

[N29]: We do not understand the question. This lack of correlation was with the rainfall values and not with aridity indexes.

[N30]: Yes. We are speaking only about the High Density (H.D.) area.

[N31]: OK. Done

[N32]: OK. Done.

 [N33]: OK. Done

[N34]: OK. Done along the whole text.

[N35]: OK. Pdf files are readable.

[N36]: OK. Done

[N37]: OK. We delete this part of the sentence.

[N38]: We were unable to obtain a better focus of this picture. Anyway, we think that the lizard is focused and only some plants are out of focus. We add a sentence to show that the lizard is looking at the infrutescence.

[N39]: You are right, we only show that there is a significant relationship between these two data, the two-years previous dispersal and the present plant density.

[N40]: Apparently, elaiosomes can be considered an alternative food resource rather than an essential part of the ant’s diet. The elaiosomes contain oleic acids and other compounds that trigger the removal behaviour of ants. Carroll & Janzen (1973) consider the elaiosomes as “dead insect analogues”. That is, as a mimic of ant’s invertebrate prey. We include a recent paper of this topic:

Caut, S., Jowers, M.J., Cerda, X. & Boulay, R.R. (2013). Questioning the mutual benefits of myrmecochory: a stable isotope-based experimen- tal approach. Ecological Entomology, 38, 390–399.

In any case, revising the sentence, we think that it is out of the lizard-plant interaction and we decided to delete it.

[N41]: We included the reference 47 because in that paper Vogel described this kind of aggressive interactions between ants and lizards, even with myrmecophagous lizard species.

[N42]: Yes, this is the question. Ants are habitual prey items of the Balearic lizard, but not during frugivorous behaviour in Helicodiceros muscivorus.

[N43]: You are right. Our paper from 2016 does not say nothing about the role of fruits of the dead horse arum as a water resource. We cited that paper because it is the largest study of feeding ecology of the Balearic lizard. Perhaps it is better to delete here that citation and to add the word “perhaps” in the sentence, to show that really, we present only a hypothesis about this supposed role of fruits.

[N44]: Yes, and at the low-density area. We add both in the text.

[N45]: You are right, the conclusion is speculative. We delete the sentence from this paragraph.

[N46]: We agree with your viewpoint. We do not have enough information about terrestrial invertebrate abundances. Thus, we modify the sentence, and we show it as a tentative hypothesis.

 [N47]: OK.

[N48]: OK.

[N49]: From several studies, not only from Aire Island, but from other locations in Western Mediterranean, the dead horse arum in many places is associated or in the close proximity to breeding colonies of seagulls. The current explanation is that the accumulation of organic matter favours the presence of bowflies, as main pollinators. Thus, this is a general comment related with the general situation of this plant species and not the description of its distribution in Aire Island.

[N50]: Really, the term “legitimate seed disperser” is widely employed in several papers and books about seed dispersal. Anyway, from our point of view, it is not important to maintain this adjective. We delete it. Answering the second question, in the remaining plant species from which we have fleshy nutritious fruits, we think that even better that the dead horse arum (see, for example, the case of Juniperus phoenicea fruits in Pérez-Cembranos & Pérez-Mellado, 2022), in those plant species, the Baleraric lizard is a seed disperser and, in some cases, the main seed disperser.

Pérez-Cembranos, A. & Pérez-Mellado, V. (2022). Scat piling and strong frugivory of the Balearic lizard, Podarcis lilfordi (Günther, 1874). BMC Zoology, 7:22 https://doi.org/10.1186/s40850-022-00125-w.

Reviewer 2 Report

I have carefully read the manuscript entitled "Long-term seed dispersal within an asymmetric lizard-plant interaction" by Ana Pérez-Cembranos, Valentín Pérez-Mellado. The work is undoubtedly of great interest and has a significant background behind it. However, after reading it, I have questions and a number of comments.

The general feeling about this manuscript I can formulate as follows: the authors have previously published a number of works related to this study and unfortunately if the reader is not familiar with the previous works, it is very difficult to understand the main hypothesis and logic of the questions formulated by the authors, which they are trying to answer using this dataset. The hypothesis must be clearly stated. In the described methodology, there is no clear description of the conditions under which the censuses of lizards and plants were carried out (time, who was taken into account? all together in the transects or? and so on). If I understand correctly, the fruits of this plant are quite large and only adult exemplars can eat them, so the overall density of lizards can be high, but the percentage of adults (which really affect the dispersal of seeds) can be low. The question of the relationship between the number of seeds in the faeces of lizards and the abundance of plants remained completely non-obvious for me. If the fruits of plants were not be eaten by lizards, they also germinate. So, how is possible to estimate true lizards impact in the distribution process. Accordingly, in order to determine the impact on the dispersal of this plant by lizards, there should be a control plot in which there are no lizards and only plants is present. At the end of the manuscript, I read the phrase that the germination of seeds increases after being eaten by lizards. And may be all these issue is clear for the authors, but to the uninvolved reader, the causal relationships in this work are not always obvious. The discussion touched on every possible aspect of the ecology of this plant and the influence of bird ants and everything else that was not directly affected by this particular study. It seems to me that would be more logical to focus on the discussion of the results obtained in this section, and to build lengthy arguments in a logical order and transfer them to the introduction. The graphics and some of the illustrations in this work are not very accurate and need to revise. Fig 1.- I'm sure you can compose this picture more accurately, besides, on maps is customary to mark the scale and North. Fig. 4 – no legend for this diagram (the colours meaning). Fig 5 - the colours are very poorly matched no chance to identify some groups (2021 vs 2022, 2006 vs 2013).

My opinion, the work should undoubtedly be published, but after revision.

Author Response

Referee #2

We try to answer each of the questions:

First, the reviewer required to make a properly formulated hypothesis. From our point of view, in the last paragraph of the Introduction we have clearly formulated this hypothesis. We propose that the intensity of the dispersal would be directly related to biotic factors such as the annual abundance of plants and the abundance of the main disperser of their seeds, the lizards. Likewise, we propose the relationship between annual rainfall, as the main abiotic factor, and the intensity of dispersion.

Later, throughout the study, we show that the abundance of plants does not have a direct effect on the intensity of dispersal, nor does the abundance of dispersers.

We clarify in the Methods section the way to do line transects during the study.

Regarding the proportion of adult and juvenile lizards during line transects, the referee is right to point out that probably, only adult lizards would be involved in seed dispersal, due to the size of fruits. But there is a small proportion of juveniles in line transects done during the fruiting period and that proportion seems to be similar along the years. In addition, lizard abundance estimated by line transects, even if we include a minor proportion of juvenile individuals, would be year to year comparable during the study.

The reviewer further comments that the relationship between the number of seeds found in feaces and the abundance of plants is not obvious. From our point of view, if we take into account the low abundance of plants at the beginning of our study in 1997 and their restricted distribution in the central zone of the island, and then we see the extraordinary expansion of the plant to areas where it did not previously exist, as well as the increase in plant density from 1999, we have to admit that a seed dispersal agent is acting during the years of study. Then, you can see in Figure 2 that there is a gap between the plat distribution in High-Density area and its distribution in Western Peninsula. The reason is that the itsmus between both areas is a very rocky area with a poor vegetal cover. Thus, with no plants. Two facts are added to these results. On the one hand, during almost 25 years of field work, we have never observed any species of bird eating the fruits of the dead horse arum and there are no mammals on the island that can act as seed dispersers. On the other hand, ants are strongly attracted to the fruits of this plant, but we do not believe that their activity can explain the rapid spread of the plant throughout the island to distances of humdreds meters. Lastly, in a previous study (Pérez-Mellado et al., 2007) we obtained experimental evidence that seeds that have passed through the digestive tract of lizards have a higher probability of germination. All these results lead us to conclude that lizards are the main dispersers of dead horse arum seeds in Aire Island. We have some additional data from restricted populations of the dead horse arums in some areas of Menorca Island and in some coastal islets, where Helicodiceros muscivorus is a rare plant and, even if we know some of these small populations from many years ago, we did not observe a population expansion and density growth comparable to those observed in Aire Island. We made some comments about this question in the Discussión section.

We agree with the referee that if the fruits are not eaten by lizards, the seeds can germinate. The key question is that in all known populations of the dead horse arum in Balearic Islands and in the remaining localities of the Western Mediterranean, the density of plants in always lower and the extension of the population very restricted (see citations included in the text). Unfortunately, we did not follow another control plot in an area of Menorca Island, for example, to show how was the population expansion and growth. Regarding seed germinability, the phrase in this text came from another published work (Pérez-Mellado et al., 2007) where we showed that experimental evidence.

Then, concerning the comment of the referee about the content of the Discussion, we think that the long comment on ants and birds is justified because there are the two groups of potential seed dispersers of the plant and, as we mention above, we need to also clarify this point.

According to the suggestion of the two referees, Figure 1 is fully redrawn.

Round 2

Reviewer 2 Report

I am quite satisfied with the response, and it remains for me to thank the authors for their detailed explanations of my comments. I wish the authors to continue their interesting research in the future, perhaps you will also be able to observe dead horse arum dispersal without lizards at the control site.